# Peer review of "Consensus-Derived Quality Performance Indicators for Neuroendocrine Tumour Care"

_jcm, 2019, doi:10.3390/jcm8091455_

Round 1
Reviewer 1 Report
This manuscript describes a clinically relevant subject. It is certainly relevant for those who are involved in developing or using quality performance indicators. However, this does not apply to most readers, and I am afraid that for them the text is too elaborate and complicated. The text is well-written with respect to the use of the English language, but could be improved for those readers. I hope that the authors would consider rewriting the text with naïve readers in mind: less details, more concrete examples (for instance an example of a performance indicator in the Introduction), more explanation of terms (a list of terms with explanations in a text box would be helpful), and making it easier to follow the main lines.
Detailed comments:
1. A short explanation of terms, as indicated above, preferably with an example would be helpful for the following terms: candidate indicators, appropriateness statements, Nominal Group Technique (is explained too late in the text), curation, stand alone, parent and de novo appropriateness statements, checkbox questions, MDM review.
2. R. 76-81: I do not fully comprehend these sentences. "Health system that cares for each individual patient": This terminology seems to point at hospitals or hospital departments, but it appears to refer to the different health systems of countries. Why is it indicated that the countries have a "predominantly socialized health system"?
3. The numbering is not always clear. For instance, are the "133 candidate QPIs" (text below Figure 1) comprehensible from the figure?
4. R. 133-134, "Candidate indicators were then 133 converted into appropriateness statements": Why was conversion necessary and how was it done?
5. R. 152-155: Why were "separate positive and negative weighted averages calculated for each statement"? Why was "each weighted average influenced by the number of respondents"?
6. R. 169-171: The numbers can be found in the supplementary file (if a row sumscore is added) and do not need to be repeated in the text. The numbers in r. 177-179 can also be placed in the supplementary file).
7. I doubt whether Figure 2 is useful. A bulleted list of simple rules would be more useful in my view, possibly added with a list of the eight statements. The supplementary file 7 seems to present the same information (It is not explained what the red bars mean, and the letters are too small to be readable). This file could also be deleted, but at least one of the two should be chosen.
8. R. 201-203, "to discuss 20 appropriate statements as shown in Figure 2 by colored dots (the three core statements not 202 shown)": It should be "the 17 statements indicated by colored dots + 3 core statements. What are the core statements, why are they called "core statements" and why are they not in the figure?
9. An example where you lose me as a reader – and possibly many later readers – is the passage from Table 1 to Figure 3. I would guess that the experts made a choice from the statements mentioned in the table, but the statements in the table and the figure are totally different.
To summarize: Please, make the text shorter, with fewer details and easier to follow, and take in rewriting the position of a reader who is not familiar with and not highly interested in QPIs. You have to draw their interest.
Reviewer 2 Report
This is an nice manuscript focusing on the role of quality performance indicators (QPIs) to improve quality of care in Neuroendocrine Tumors (NETs).
Authors apply a rigorous and novel methodology to this rare and heterogeneous kind of cancer, making this work novel and interesting for physicians dealing with NETs.
Furthermore, the paper is well-written and clear for the reader.
I have only few minor comments which I would appreciate to be considered when reviewing the manuscript:
the need of somatostatin receptor functional imaging was not selected to be included in the final consensus. In my opinion, this point deserves a specific comment. Progression-free survival was not analyzed/discussed in the consensus. Since it is well-known that this is an effective endpoint when approaching NET clinical outcome, a comment on this issue should be added in the discussion. Authors could try to translate the final consensus-derived NET QPIs summarized in table 2 into a figure reporting a proposal on suggested behavior for physicians to be applied in the daily clinical practice...I mean, when facing with a NET patient: seek for primary site (if not, perform somatostatin receptor imaging / additional radiology...) > check proliferative index is available (otherwise ask for histological revision) > exclude M1 disease (with particular effort on looking for distant extra-hepatic metastases which are known to worsen patients' clinical outcome - this issue also deserves a comment) etc etc...
Round 2
Reviewer 1 Report
The manuscript has certainly been improved, but there are still too many places where I lose sight of the main lines:
The main procedures, especially a short description of the various phases, should be placed in the - beginning of the Methods section. My proposal: The first phase (round 0) aimed at formulating as many statements related to the treatment of NET that were seen as relevant. Next, some experts reformulated the items, such as to make them clear, unambiguous and appropriate for the evaluation ("curation") in round 1. Round 1 was aimed at reducing the number of statements to a manageable number by letting them rated by a large number of participants with respect to measurability and importance. In the last phase – round 2 – the smaller set of statements was one more time evaluated by a smaller group of experts, mainly consisting of medical doctors who will later have most to do with the quality performance indicators.
[I got only later in the manuscript some idea of the different aims of round 1 and 2. I hope to have correctly described these differences. Would it not be easier to number the rounds from 1 to 3, instead of from 0 to 2?]
R. 76-79, "For example, while detection of carcinoid heart disease is highly relevant for people with NETs with carcinoid syndrome, only a small fraction of people with NETs are affected thus questioning its value as a general indicator of quality care":
Though I myself have asked for examples, I do not totally understand this sentence. "Detection of carcinoid heart disease .. in people with carcinoid syndrome". What does this mean? Detection of heart disease in people with cardiac symptoms? I would begin this sentence as "Some diseases are relevant in people with NETs, but are very rare, thus questioning its value as a general indicator of quality care. An example is …".
The following title of paragraph 2.2 (r. 101) is easier to understand: "The generation of candidate statements" R. 111, Title 2.3: What is the difference between curation and the conversion into appropriateness statements? A simple title would be "The conversion of candidate statements into appropriate statements". R. 123, "… noted to be included": What does it mean? R. 130, Title 2.4: Why not simply "Online survey"? R. 132-134, "An introductory … the survey": A superfluous sentence. Why does the reader needs to know all these details? R. 142-150: I now understand why you applied this procedure, but I do not understand – even after reading these lines and the Supplementary file 5 several times – the procedure itself. Can you, please, reconsider this procedure and how you could explain it simpler. Text under Figure 1, after "106 appropriate statements": Clearer and simpler seems the following continuation "In round 1 these statements were evaluated, using the RAND/UCLA appropriateness method. The participants rated the importance and measurability of each statement as indicators of care quality in an online survey, which lead to a reduction of up to twenty statements. In round 2, a small group of experts discussed these twenty statements and - after some adjustments – rated them online, leading to a final list of ten QPIs". R. 169-172: How can 133+42-28-10=137 be 62+44=106. Is the report of all these numbers necessary. If you think so, they must be understandable. R. 173-176: The numbers can be find in the figure, and there is no need to repeat them here. R. 175-176, ".. seven parent statements were created de novo to allow comparison of 14 additional sub-statements": Why not simply say that seven new parent statements were created" (See also Figure 1). I do not understand the reason why you have added these new statements. R. 182-183, "(after removal of the three core statements)": Why were the removed? Because there was no need to evaluate them? If so, I would say "(without the three core statements, for which an evaluation was not needed)". R. 190, "(n=59)": Can better be deleted. It is uncertain whether this number refer to participants or statements. R. 203-204: Please give an example of an old phrase and a revised one. Table 1: What do "Up-ranked statements" and "Aspirational indicators" mean?
